# Meta-2OM: A multi-classifier meta-model for the accurate prediction of RNA 2′-O-methylation sites in human RNA

**Md. Harun-Or-Roshid**[1], **Nhat Truong Pham**[2], **Balachandran Manavalan**[2]\*, **Hiroyuki Kurata**[1]\*

**1** Department of Bioscience and Bioinformatics, Kyushu Institute of Technology, Iizuka, Fukuoka, Japan,
**2** Department of Integrative Biotechnology, College of Biotechnology and Bioengineering, Sungkyunkwan University, Suwon, Republic of Korea

\* kurata@bio.kyutech.ac.jp (HK); bala2022@skku.edu (BM)

**Data Availability Statement:** All relevant data are within the paper and its Supporting Information files.

## Abstract

2′-O-methylation (2-OM or Nm) is a widespread RNA modification observed in various RNA types like tRNA, mRNA, rRNA, miRNA, piRNA, and snRNA, which plays a crucial role in several biological functional mechanisms and innate immunity. To comprehend its modification mechanisms and potential epigenetic regulation, it is necessary to accurately identify 2-OM sites. However, biological experiments can be tedious, time-consuming, and expensive. Furthermore, currently available computational methods face challenges due to inadequate datasets and limited classification capabilities. To address these challenges, we proposed Meta-2OM, a cutting-edge predictor that can accurately identify 2-OM sites in human RNA. In brief, we applied a meta-learning approach that considered eight conventional machine learning algorithms, including tree-based classifiers and decision boundary-based classifiers, and eighteen different feature encoding algorithms that cover physicochemical, compositional, position-specific and natural language processing information. The predicted probabilities of 2-OM sites from the baseline models are then combined and trained using logistic regression to generate the final prediction. Consequently, Meta-2OM achieved excellent performance in both 5-fold cross-validation training and independent testing, outperforming all existing state-of-the-art methods. Specifically, on the independent test set, Meta-2OM achieved an overall accuracy of 0.870, sensitivity of 0.836, specificity of 0.904, and Matthew's correlation coefficient of 0.743. To facilitate its use, a user-friendly web server and standalone program have been developed and freely available at http://kurata35.bio.kyutech.ac.jp/Meta-2OM and https://github.com/kuratahiroyuki/Meta-2OM.

## Introduction

Post-transcriptional chemical modification of ribonucleic acid (RNA) plays significant functions in cellular regulation and biological information in all three phylogenetic domains (i.e. eukaryotes, bacteria, and archaea) of the life [1, 2]. Among various RNA alterations, nucleotide

**Funding:** This work was supported by a Grant-in-Aid for Scientific Research (B) (22H03688 to H.K.) from the Japan Society for the Promotion of Science (JSPS), Japan and the National Research Foundation of Korea (NRF) funded by the Ministry of Science and ICT (2021R1A2C1014338 to B. M.), Republic of Korea, supported by the Department of Integrative Biotechnology, Sungkyunkwan University (SKKU) and the BK21 FOUR Project.

modification stands out for its evolutionarily conserved nature within the epitranscriptome [3]. With the advances in genomics and molecular biology, approximately 300 RNA modification types have been identified, playing a crucial role in regulating transcriptional processes [3–5]. One prevalent modification is 2'-*O*-methylation (2-OM), where a methyl group (-CH$_3$) is added to the $2'$ hydroxyl (-OH) of the ribose moiety of a nucleoside. It is also known as the Nm modification of RNA where 'N' stands for any nucleotide. 2-OM nucleotides are predominantly found in ribosomal RNAs (rRNAs), small nuclear/nucleolar RNAs (snRNAs), transfer RNAs (tRNAs), piwi-interacting RNAs (piRNAs), microRNAs (miRNAs), and messenger RNAs (mRNAs) [6–12]. The 2-OM modification alters the RNA activities in different, epigenetic ways such as translation regulation and secondary structure stability, and provides a molecular signature for the discrimination of self and non-self mRNAs [13–16]. Furthermore, 2-OM links to various human diseases [3], including Prader–Willi syndrome, asthma, Alzheimer's disease [13], and breast cancer [17]. Interestingly, 2-OM is also being investigated as a potential target for drugs that could prevent the early innate immune evasion of the SARS-Coronavirus 2 viral RNA [18, 19]. In light of these, accurate detection of 2-OM sites is essential for developing a deeper understanding of their functional implications.

Several experimental methods have been developed for identifying 2-OM sites, both before and after the era of high-throughput sequencing technologies. Some studies have proposed identification based on molecular approaches, with the goal of precisely identifying Nm sites in the ncRNA [19]. Krogh et al. [20], Erales et al. [21], Sharma et al. [22], and Zhou et al. [23] proposed several *in-vitro* techniques to detect the 2-OM sites, but these techniques were only partially effective and were specifically sensitive to p53. RibOxi-seq [24] is another *in-vitro* method that can be used to discriminate 2-OM modifications in rRNA. Dai et al. [12] proposed a high-throughput experimental method named Nm-seq, which is used to detect Nm-modification sites in mRNA. Experimental methods can detect 2-OM sites in the transcriptome, but they are time-consuming, labor-exhaustive, and expensive. Computational methods, on the other hand, are efficient and effective, and they play an important role in many areas of bioinformatics. For example, in silico techniques are being used rapidly in research on disease-gene interactions [25, 26], protein structure prediction [27], peptide therapeutic function, gene editing experiments [28], meaningful pattern detection [29], and drug repurposing [30, 31]. Previously, researchers have been proposed a few computational models for predicting the 2-OM sites based on single machine learning (ML) and deep learning (DL) approaches [32–39].

A summary of these methods is provided in Table 1. Chen et al. [39] developed an support vector machine (SVM)-based prediction model trained on the benchmark dataset of 147 positive and 147 negative samples of Am modifications. Huang et al.'s [34], iRNA-PseKNC (2methyl) [35], iRNA-2OM [37], and NmRF [32] were also built using the same dataset as proposed by Chen et al. Although these methods achieved high prediction performance, they did not generalize well to other types of 2-OM (Gm, Cm, Um) sites. Deep-2'-O-Me [38] is a convolutional neural Network (CNN) with word2vec feature encodings that achieved area under the receiver operating characteristic curves (AUC) of 90% on an independent test for both balanced and unbalanced datasets. NmSEER V2.0 [36] is an updated version of NmSEER [40] that uses random forest (RF) with one-hot, position-specific nucleotide sequence profile (PSNSP), and *K*-nucleotide frequencies (KNF) feature encoding to achieve an AUC of 86.2% for predicting Nm sites. DeepOMe [33] is a CNN-bidirectional long short-term memory (BLSTM) hybrid method that achieved an accuracy (ACC) of 0.956 and AUC of 0.998 on an independent test dataset. NmRF [32] is an RF-based method trained on optimal mixed features to identify 2-OM sites in multiple species. It achieved ACCs of 0.890 and 0.939 for humans and yeast, respectively. The ML-oriented latest method, i2OM [41] is an SVM- and

**Table 1. Summary of available methods to predict 2-OM sites.**

| Year | Method | Algorithm | Feature techniques | Sample size[a] | Testing method | ACC | Genomes |
|---|---|---|---|---|---|---|---|
| 2023 | H2Opred [42] | CNN, Bi-GRU | 19 encodings | 6091/6091 | 5-fold CV and Independent test | 0.858 0.910 | Human |
| 2023 | i2OM [41] | SVM, XGBoost | K-mer, NCP, ANF | 6091/6091 | 5-fold CV and Independent test | 0.863 0.843 | Human |
| 2022 | NmRF [32] | RF | mixed | 147/147 | 10-fold CV and Independent test | 0.891 0.939 | Human, Yeast |
| 2021 | DeepOMe [33] | CNN, BLSTM | One-hot | 3052 | 10-fold CV | 0.956 | Human |
| 2020 | Huang et al. [34] | SVM | - | 147/147 | 10-fold CV | 0.765 | Human |
| 2019 | iRNA-PseKNC (2methyl) [35] | CNN | One-hot | 147/147 | 5-fold CV | 0.983 | Human |
|  | NmSEER V2.0 [36] | RF | One-hot, PSDSP, KNF | 1989/1989 | 5-fold CV | 0.862 | Human |
| 2018 | iRNA-2OM [37] | SVM | NCP, NC, PseKNC | 147/147 | 5-fold CV | 0.979 | Human |
|  | Deep-2'-O-Me [38] | CNN | word2vec | -/- | Independent test | AUC-0.900 | Human |
| 2016 | Chen et al. [39] | SVM | NCP, NC | 147/147 | Jacknife CV | 0.956 | Human |

[a]Sample size: positive /negative dataset

the eXtreme gradient boosting (XGB)-based method that can predict the four types of nucleotide modification in 2-OM sites. It achieved AUCs of 0.920, 0.869, 0.933, and 0.936 for Am, Um, Gm, and Cm, respectively, on an independent test dataset. The DL-based method H2Opred [42] was designed to predict the 2-OM modification for human RNAs. The H2Opred method was trained using both generic and nucleotide-specific datasets. Surprisingly, the generic model outperformed the nucleotide-specific models, which achieved the AUCs of 0.954, 0.949, 0.958, and 0.928 for Am, Cm, Gm, and Gm, respectively, on the independent datasets.

Most previously published methods for predicting 2-OM sites are limited in scope, such as only predicting 2-OM sites in a single type of RNA dataset (e.g., mRNA or rRNA), or a specific type of nucleotide modification sites, which are built based on smaller training datasets. In this study, we developed a more powerful integrated predictor Meta-2OM that predicts all 2-OM sites (Am, Cm, Gm, Um) in human epitranscriptome sequencing data. Fig 1 illustrates the overall framework of Meta-2OM, which is based on a meta-learning approach that explores eight ML classifiers (RF, SVM, XGB, light gradient boosting machine (LGBM), Catboost classifier (CBC), Naïve Bayes (NB), K-nearest neighbor (KNN), Logistic regression (LR)) with 18 diverse RNA sequence-based feature encoding methods. Subsequently, the predicted probabilities of 2-OM sites from multiple baseline models were concatenated and trained with LR to generate the meta-classifier. This approach allows Meta-2OM to learn from the strengths and weaknesses of each model, resulting in a more accurate and robust prediction. Meta-2OM was evaluated on an independent test set and performed slightly better than the state-of-the-art methods, demonstrating its effectiveness as the most effective predictor of 2-OM sites to date.

## Materials and methods

### Benchmark dataset construction

In this study, we utilized a benchmark dataset specifically designed for the development and evaluation of predictive models for 2-OM RNA modification sites in humans (*Homo sapiens*) RNA sequences. The dataset, originally constructed by Yang et al. [41] was derived from sequence data from RMBase v2.0 [43] and experimental datasets from Nm-seq (GSE90164) [12]. It comprised a total of 7,597 positive samples (2-OM or Nm sites) across various RNA types, including tRNAs, rRNAs, scRNAs, snRNAs, snoRNAs, scaRNAs, lincRNAs, protein-coding genes, and pseudogenes. Each sample was represented by a sequence of 41 nucleotide

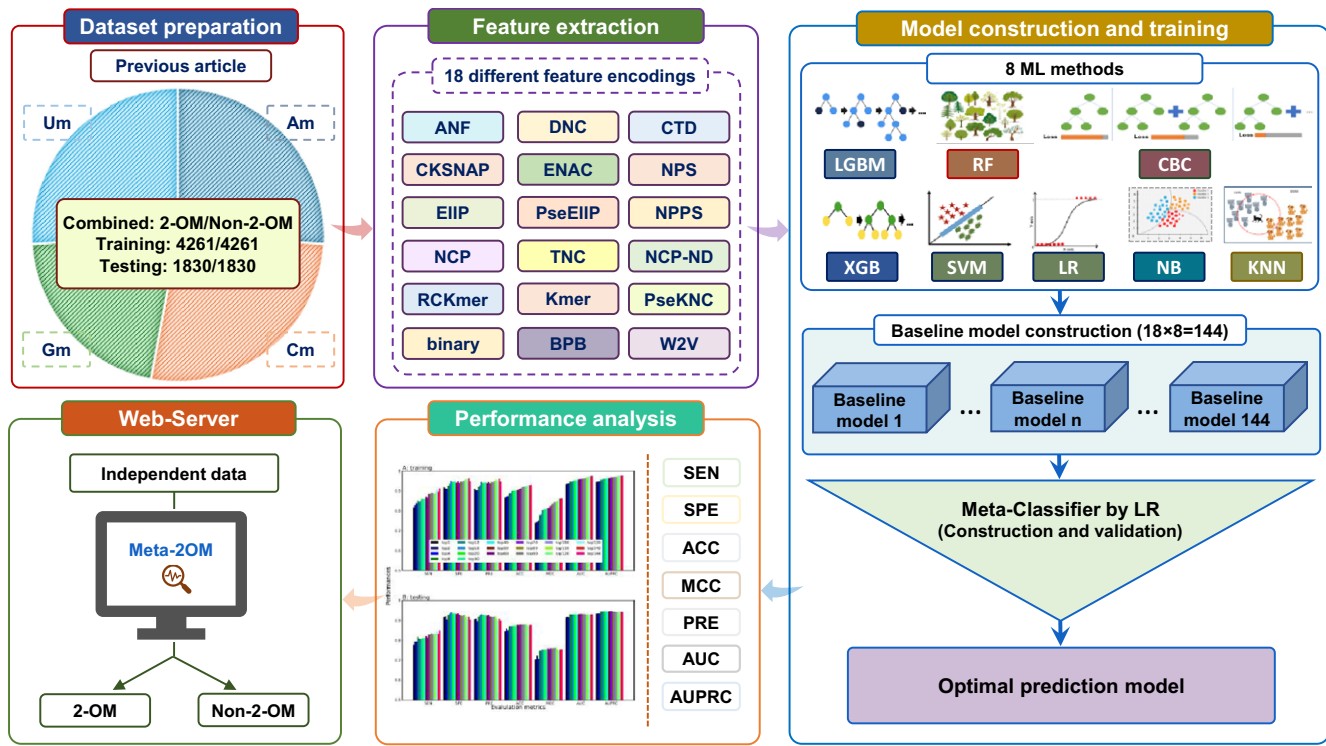

**Fig 1. The development workflow of Meta-2OM.** It consists of five major steps: (i) Data collection and preparation: collect and prepare 2-OM benchmark datasets from databases and split them into the training and independent test datasets. (ii) Feature extraction: extracting features using 18 different encoding methods. (iii) Model construction and training: constructing the baseline classifiers using eight different ML methods and meta-classifiers using LR for final prediction. (iv) Performance analysis: analyze and compare the performance of the meta-classifier, and (v) Web server construction: constructing a user-friendly web application.

base pairs (bp), centered around the modification site. To remove sequence redundancy, the CD-HIT [44] was employed with a threshold of 0.80, yielding 6,091 positive samples. The dataset also included 21,520 negative samples, representing non-2-OM or non-Nm sites. They balanced the dataset to a 1:1 ratio of positive to negative samples to minimize training bias associated with imbalanced datasets. As a result, they developed four modification-specific (Am, Cm, Gm, Um) and generic (Nm) dataset, which were then divided into training and testing sets. The statistical summary of these datasets is provided in Table 2.

## Encoding and feature extraction strategy

Feature extraction methods are crucial for developing sequence-based ML methods to detect not only RNA modification sites [45, 46], but also other function prediction problems. Therefore, we explore 18 different feature encoding methods, which are briefly described below.

**Table 2. Description of benchmark datasets.**

| Datasets (*H. sapiens*) | Total sample | Training | | Independent | |
|---|---|---|---|---|---|
| | | Positive | Negative | Positive | Negative |
| Nm (2-OM) | 12,182 | 4,261 | 4,261 | 1,830 | 1,830 |
| Am | 3,110 | 1,088 | 1,088 | 467 | 467 |
| Cm | 3,274 | 1,138 | 1,138 | 498 | 498 |
| Gm | 2,620 | 916 | 916 | 394 | 394 |
| Um | 3,196 | 1,118 | 1,118 | 480 | 480 |

**Binary.** Binary encoding converts a single nucleotide (A, C, G, U) into a four-dimensional binary vector (0, 1) [47]. For example, the nucleotides A, C, G, and U of are represented as (1, 0, 0, 0), (0, 1, 0, 0), (0, 0, 1, 0) and (0, 0, 0, 1), respectively. Therefore, a 41-bp nucleotide sequence can be represented as a 164-dimensional feature vector.

**Enhanced Nucleic Acid Composition (ENAC).** ENAC encodes sequences based on the local nucleic acid composition (NAC) with a fixed-length window of size $k$, starting at the 5′ end and moving to the 3′ end. The default $k$ is 5 [48]. For a 41-bp sequence, ENAC produces 148 dimensional feature vector, which is defined as $E = (b_1, b_2, \ldots, b_n)$ and

$$b_j = \frac{N_i}{k}, \ i \in \{A, C, G, U\} \tag{1}$$

where $N_i$ represents the number of $i$-th nucleotides in $j$-th window and $n$ is (sequence length $- k + 1$).

**Nucleotide Chemical Property (NCP).** NCP converts RNA nucleotides into three-dimensional vectors based on ring structure, functional group, and hydrogen bond properties. NCP encodes the nucleotides, A, C, G, and U as (1, 1, 1), (0, 1, 0), (1, 0, 0), and (0, 0, 1), respectively. For a given 41-bp RNA sequence, NCP produces a 3×41 (= 123)-dimensional feature vector.

**Accumulated Nucleotide Frequency (ANF).** ANF extracts numeric information from nucleotide sequences by analyzing the distribution of nucleotide types in the RNA [49]. The mathematical function of the ANF feature for a specific nucleotide $n_i$ at the $i$-th position in the RNA sequence is as follows:

$$d_i = \frac{1}{|N_i|} \sum_{j=1}^{i} f(n_j), \ f(n_j) = \begin{cases} 1 \ if \ n_j = n_i \\ 0, otherwise \end{cases} \tag{2}$$

where $|N_i|$ represents the length string up to the $i$-th position $\{n_1, n_2, \ldots, n_i\}$ in the sequence. Therefore, the ANF converts 41-bp RNA sequences into 41-dimensional feature vectors.

**Electron-Ion Interaction Pseudopotentials (EIIP).** EIIP encoding converts RNA sequence data into numeric feature vectors by replacing each nucleotide with its corresponding electron energy. The electron energies for A, C, G, and U, are 0.1260, 0.1340, 0.0806 and 0.1335, respectively [50]. This method generates a 64-dimensional feature vector for a sequence. It is computed as follows:

$$D = [EIIP_{AAA} \times f_{AAA}, EIIP_{AAC} \times f_{AAC}, EIIP_{AAG} \times f_{AAG}, \ldots, EIIP_{UUU} \times f_{UUU}] \tag{3}$$

where $EIIP_{mno} = EIIP_m + EIIP_n + EIIP_o$, and $m, n, o \in \{A, C, G, U\}$; $f_{mno}$ represents the trinucleotide frequency.

**Di-Nucleotide Composition (DNC).** DNC encoding generates a 16-dimensional feature vector for an RNA sequence by counting the frequencies of all possible dinucleotide combinations. DNC can be extracted as:

$$f(m, n) = \frac{K_{m,n}}{K - 1}, \ m, n \in \{A, C, G, U\} \tag{4}$$

where $K_{m,n}$ is the dinucleotide combination frequencies by $m$ and $n$ nucleotides.

**Tri-Nucleotide Composition (TNC).** TNC encoding generates a 64-dimensional feature vector for an RNA sequence by counting the frequencies of all possible trinucleotide combinations. TNC can be expressed as:

$$f(m, n, o) = \frac{K_{m,n,o}}{K - 2}, \ m, n, o \in \{A, C, G, U\} \tag{5}$$

where $K_{m,n,o}$ represent the frequencies of $m$, $n$, and $o$ trinucleotide combinations, and $m$, $n$, and $o$ can make any set among 'AAA', 'AGA', 'ACA', ..., and 'UUU'.

**Composition of $k$-spaced Nucleic Acid Pairs (CKSNAP).** CKSNAP encoding calculates the frequencies of $k$-spaced nucleotides pairs in an RNA sequence, where $k$ (0, 1, 2, 3, 4, 5) is the distance between nucleotides in the pair [51]. If we consider $k = 0$, then we get 16 $k$-spaced nucleotide pairs: 'AA', 'AC', 'AG', 'AU', 'CA', 'CC', 'CG', 'CU', 'GA', 'GC', 'GG', 'GU', 'UA', 'UC', 'UG', 'UU. The CKSNAP encoding can be expressed as follows:

$$\left( \frac{N_{AA}}{N_{total}}, \frac{N_{AC}}{N_{total}}, \frac{N_{AG}}{N_{total}}, \frac{N_{AG}}{N_{total}}, \frac{N_{CA}}{N_{total}}, \cdots, \frac{N_{UU}}{N_{total}} \right)_{16} \tag{6}$$

where $N_{AA}$ represents the total number of nucleotide pairs in the entire RNA sequences and $N_{total}$ is the total number of pairs in the RNA sequences with the gap space $k$.

**Reverse Complement Kmer (RCKmer).** RCKmer is a concise variant of Kmer that works by removing reverse complement pairs. In Kmers, at $k = 2$, we obtain 16 Kmer nucleotides (i.e., 'AA', 'AC', 'AG', 'AU', 'CA', 'CC', 'CG', 'CU', 'GA', 'GC', 'GG', 'GU', 'UA', 'UC', 'UG', 'UU'). However, after removing reverse Kmer (e.g., 'UU' is the reverse complement of 'AA'), we are left with 10 unique Kmers: 'AA', 'AC', 'AG', 'AU', 'CA', 'CC', 'CG', 'GA', 'GC' and 'UA'. These 10 Kmers can be then used to create a feature vector.

**Pseudo Electron-Ion Interaction Pseudopotentials (PseEIIP).** PseEIIP is a feature encoding method that calculates the average values across trinucleotide sequences to generate a 64-dimensional feature vector, consistent with the dimensionality of the original EIIP model. By focusing on the mean EIIP values, PseEIIP captures the overall electron-ion interaction tendencies within trinucleotide segments, providing a nuanced yet comprehensive feature set.

**BPB approach.** The Bi-profile Bayes (BPB) feature encoding is a technique that encodes the sequence data using the posterior probability [52]. To calculate the posterior probability of each position of the sequence samples, we consider the following Baye's theorem:

$$P(f^+|S) = \frac{P(S|f^+)P(f^+)}{P(S)} \tag{7}$$

$$P(f^-|S) = \frac{P(S|f^-)P(f^-)}{P(S)} \tag{8}$$

where $S = \{s_1, s_2, \cdots, S_L\}$ is the sequence sample, $f^+$ is the positive dataset, $f^-$ is the negative dataset, $P(f^+)$ is the prior probability, $P(f^+|S) = \{p_1^+, p_2^+, \cdots, p_L^+\}$ and $P(f^-|S) = \{p_1^-, p_2^-, \cdots, p_L^-\}$ are the posterior probabilities for positive and negative samples, respectively. $L$ is the length of sequences. The BPB method generates a numeric vector for a sequence sample $\mathbf{D}_{BPB}$ with dimension $L \times 2$ by combining their bi-profiles posterior probabilities.

**CTD approach.** The CTD feature encoding method is based on three sequence information descriptors: C (nucleotide composition), T (nucleotide transition), and D (nucleotide distribution) [53]. Nucleotide composition defines the percentage of each nucleotide within a sequence. Nucleotide transition describes the frequency percentage of transition between the four nucleotides at adjacent positions. The third descriptor, nucleotide distribution was calculated based on five relative positions: 0 (first one), 25%, 50%, 75%, and 100% (last one) for each nucleotide within the sequence. This method results in the generation of a 30-D feature vector.

**Kmer.** The Kmer encoding method constructs a feature vector based on the nucleic acid composition within segments of $k$ nucleotides. The value of $k$ can vary corresponding mono-, di-, tri-, tetra-, and pentanucleotides. This technique can generate a $4^k$ dimensional feature

vectors. For instance, when $k = 2$, the method generates $4^k = 16$ dimensional feature vector for a given RNA sequence by counting the frequencies of all possible dinucleotide combinations. The mathematical representation is as follows:

$$f(m, n) = \frac{K_{m,n}}{K - 1}, \ m, n \in \{A, C, G, U\} \tag{9}$$

where $K_{m,n}$ is the dinucleotide combination frequencies by $m$ and $n$ nucleotides. Here, we set $k$ to 4.

**NCP-ND approach.** The nucleotide chemical properties and nucleotide distribution (NCP-ND) encoding utilizes both the chemical property of nucleotides and their specific distribution at given positions to generate a combined features [54]. In the NCP component, nucleotides $A$, $C$, $G$, and $U$ are encoded as $(1, 1, 1)$, $(0, 1, 0)$, $(1, 0, 0)$, and $(0, 0, 1)$, respectively, based on the properties of ring structure, hydrogen bond, and functional group. The ND calculates the distribution $d_i$ of the $i$-th nucleotide $m_i$ as follows:

$$d_i = \frac{1}{i} \sum_{j=1}^{L} f(m_j), \quad f(q) = \begin{cases} 1, & if \ m_j = q \\ 0, & otherwise \end{cases} \tag{10}$$

where $q \in \{A, C, G, U\}$ and $L$ is the sequence length. Therefore, each of the sample sequences becomes an $L \times 4$ dimensional numerical vector $\mathbf{D}_{\text{NCP\_ND}}$.

**NPS approach.** The nucleotide pair spectrum (NPS) is a sequence encoding method that generates an occurrence frequency vector of an RNA sequence by counting the $k$-spaced nucleotide pairs [55]. The $k$-spaced nucleotide pair $n_1\{k\}n_2$ indicates the nucleotide pair of $n_1$ and $n_2$ excluding $k$-spaces. The occurrence frequency can be generated as follows:

$$F_{n_1\{k\}n_2} = \frac{C(n_1\{k\}n_2)}{L - k - 1} \tag{11}$$

where $C(n_1\{k\}n_2)$ is the number of nucleotide pair $n_1\{k\}n_2$ in a sequence window with length $L - k - 1$. The parameter $k$ comes from 0 to $d_{max}$, and $d_{max} = 3$ and each of the RNA sequences can be converted as a numeric vector $\mathbf{D}_{NPS}$ with a dimension of $4 \times 4 \times (d_{max} + 1)$.

**NPPS approach.** The nucleotide pair position specificity (NPPS) encoding method quantifies the statistical information based on the position-specific single nucleotide and $k$-spaced nucleotide pairs [56]. For a given sequence, it calculates the frequency matrix as follows:

$$F_s^+ = \begin{bmatrix} f_{S(A,1)}^+ & f_{S(A,2)}^+ & \cdots & f_{S(A,L)}^+ \\ f_{S(C,1)}^+ & f_{S(C,2)}^+ & \cdots & f_{S(C,L)}^+ \\ f_{S(G,1)}^+ & f_{S(G,2)}^+ & \cdots & f_{S(G,L)}^+ \\ f_{S(U,1)}^+ & f_{S(U,2)}^+ & \cdots & f_{S(U,L)}^+ \end{bmatrix} \tag{12}$$

$$F_d^+ = \begin{bmatrix} f_{d(AA,1)}^+ & f_{d(AA,2)}^+ & \cdots & f_{d(AA,L-k-1)}^+ \\ f_{d(AC,1)}^+ & f_{d(AC,2)}^+ & \cdots & f_{d(AC,L-k-1)}^+ \\ \vdots & \vdots & \ddots & \vdots \\ f_{d(UU,1)}^+ & f_{d(UU,2)}^+ & \cdots & f_{d(UU,L-k-1)}^+ \end{bmatrix} \tag{13}$$

where $F_s^+$ is the $(4 \times L)$ dimensional probability matrix of the single nucleotides occurring in the specific position of positive data samples, $F_d^+$ is $(16 \times (L - k - 1))$ dimensional probability matrix of the $k$-spaced nucleotide pairs occurring in the specific position of positive data sample.

Similarly, the probability matrix $F_s^-$ and $F_d^-$ are generated for the negative data samples. Then the conditional probability between a single nucleotide ($A$) and $k$-spaced nucleotide pair ($AG$) can be calculated as:

$$p_i^+ = \frac{f_{d(AG,i)}^+}{f_{s(G,i+k)}^+} \text{ and } p_i^- = \frac{f_{d(AG,i)}^-}{f_{s(G,i+k)}^-} \tag{14}$$

Finally, the NPPS encoding converts a sequence to an ($L-k-1$) dimensional vector:

$$\boldsymbol{D}_{NPPS} = P^+ - P^- \tag{15}$$

where $P^+ = [p_1^+, \cdots, p_{L-k-1}^+]$ and $P^- = [p_1^-, \cdots, p_{L-k-1}^-]$. In this study, we set k to 0.

**PseKNC approach.** Pseudo $k$-nucleotide component (PseKNC) [57] converts the sequence data into the feature vector using the local and global information of RNAs as follows:

$$D = [d_1, d_2, \cdots, d_{4^k}, d_{4^k+1}, \cdots, d_{4^k+\lambda}]^T \tag{16}$$

where

$$d_u = \begin{cases} \dfrac{f_u}{\sum_{i=1}^{4^k} f_i + w \sum_{j=1}^{\lambda} \theta_j} & (1 \leq u \leq 4^k) \\[4mm] \dfrac{w\theta_{u-4^k}}{\sum_{i=1}^{4^k} f_i + w \sum_{j=1}^{\lambda} \theta_j} & (4^k \leq u \leq 4^k + \lambda) \end{cases} \tag{17}$$

$d_u$ ($u = 1,2,\cdots,4^k$) is the occurrence frequency of $k$-tuple nucleotide compositions, $w$ is the weight factor, $\lambda$ is the number of the counted compositions correlated along RNA sequences, and

$$\theta_j = \frac{1}{L-j-1} \sum_{i=1}^{L-j-1} \Theta(R_i R_{i+1}, R_{i+j} R_{i+j+1}), \quad (j = 1, 2, \ldots, \lambda; \lambda < L) \tag{18}$$

$$\Theta\left(R_i R_{i+1}, R_{i+j} R_{i+j+1}\right) = \frac{1}{\mu} \sum_{v=1}^{\mu} [P_v(R_i R_{i+1}) - P_v(R_{i+j} R_{i+j+1})]^2 \tag{19}$$

where $\mu$ is the number of RNA physiochemical properties used, $R_i R_{i+1}$ is the $i$-th position dinucleotide, and $P_v(R_i R_{i+1})$ is the corresponding standardized value of $v$-th RNA local structural property.

**Word2vec.** Word2vec (W2V), an essential feature embedding method based on natural language processing (NLP), is extensively employed in text data analysis and has proven particularly effective in various bioinformatics pattern recognition tasks involving sequence data [58–61]. W2V utilizes two algorithms: continuous bag-of-words (CBOW) and continuous skip-gram. While CBOW predicts the current words from its context, skip-gram predicts the context from the neighboring words. In this study, we employed the skip-gram algorithm to train a W2V model on 162 RNA sequences retrieved from RNAcentral [62] using the key words "Human", "Rfam", and "non-coding RNA". This resulted in 64-dimensional feature vectors for each nucleotide.

## Implemented ML classifiers

We implemented eight ML methods to develop a predictor for identifying 2-OM RNA modification sites. These methods included four tree-based algorithms (RF, XGB, LGBM, and CBC),

two decision boundary-based classifiers (SVM, LR), and other classifiers (NB, KNN). It should be noted that all ML models were implemented with the following libraries and packages, including the scikit-learn (https://scikit-learn.org/), XGBoost (https://xgboost.readthedocs.io/), CatBoost (https://catboost.ai/) and lightGBM (https://lightgbm.readthedocs.io/). A brief description of these methods is provided as follows:

**RF.** RF is one of the most popular supervised ML algorithms, which is broadly used in bioinformatics for wide-range of problems [63–69]. RF combines numerous decision trees trained on the different training samples and predicts the class of new samples based on the majority voting techniques. Thus, the RF method delivers a highly powerful classification decision.

**SVM.** SVM is a robust supervised learning algorithm that are widely used for classification problems. SVMs work by finding a hyperplane, which is a flat surface in a higher-dimensional space, that best separates the data points into two classes. To improve their ability to classify complex data, SVMs use a kernel function to transform the low-dimensional data into high-dimensional space. This makes SVMs as a powerful tool for bioinformatics applications, where the data if often high-dimensional and complex [45, 68–70].

**XGB.** The XGB is a popular ensemble-based ML algorithm that is widely used in regression and classification tasks, especially on large datasets, which is common in bioinformatics [41, 45]. XGB achieves high predictive ACC by combining the predictions of multiple individual decision trees, which are trained using a gradient boosting approach. One of the key strengths of XGB is its ability to control overfitting, which is a common problem in ML. XGB does this by using L1 (Lasso regression) and L2 (ridge regression) regularization methods to penalize the weights of the individual decision trees. This helps to reduce the complexity of the ensemble model and improve its robustness. XGB is also designed to efficiently handle sparse datasets, which are common in high-dimensional biological data. XGB's architecture is optimized for scalability and parallel processing, allowing it to perform computations on multiple cores, which significantly reduces memory usage and computational time.

**LGBM.** LGBM is a highly efficient ML algorithm that is known for its fast training times, high efficiency, low memory usage, and ability to handle large datasets without sacrificing accuracy on a variety of tasks, including classification, regression, and ranking [71]. Like XGB, LGBM is based on decision trees, but it incorporates several advanced computing techniques, such as optimizations for sparse data structures, parallel processing, multiple loss functions, regularization techniques, bagging, and early stopping to prevent overfitting. What particularly sets LGBM apart are two innovative methodologies: Gradient-based One-Side Sampling (GOSS) and Exclusive Feature Bundling (EFB). GOSS accelerates the training process by keeping the most informative instances while filtering out less important ones, resulting in more focused and efficient learning. EFB, on the other hand, improves efficiency by grouping mutually exclusive features, effectively reducing the dimensionality of the feature space without significant information loss.

## CBC

CBC represents an advanced ML-based algorithm, which is based on gradient-boosted decision trees methodology [72, 73]. This algorithm is particularly effective in handling both classification and regression tasks that incorporate categorical features. CBC employs a combination of ordered boosting, random permutations, and gradient-based optimization to achieve superior classification performance on large and complex datasets featuring categorical variables. Its robustness and efficiency have led to widespread adoption in various bioinformatics domains, notably in RNA modification identification task [74].

### NB

A NB classifier is the probabilistic ML model built based on the principles of Bayes' theorem, with the foundational assumption of conditional independence among all pairs of input features. This model is extensively applied across various bioinformatics challenges, especially sequence-based function prediction [75]. Among the different variants of Naive Bayes classifiers, we implemented here the Gaussian Naive Bayes classifier.

### KNN

The KNN algorithm is a popular ML technique used for classification and regression tasks. It relies on the idea that similar data points tend to 'K' most similar data points in the training dataset. The homogeneity of different data points is measured by the Euclidean distance method to find the neighbors. Moreover, the right choice of 'K' is crucial to better performance of the KNN method. It is an effective method in classification tasks of bioinformatics fields [76, 77].

### LR

LR is a well-known generalized linear ML technique. It is used to classify binary classification problems using datasets without multi-collinearity. LR produces a probability value that lies between 0 to 1 for each data sample, which can be used to predict the likelihood of a sample belonging to one of the two classes. This makes LR a useful technique for bioinformatics applications, such as DNA, RNA, and protein modification site prediction [78, 79]. Additionally, the search range for the hyperparameters of ML classifiers is given in the S1 Table.

### Meta-learning approach

We generated baseline models (BMs) by combining eight ML algorithms with 18 feature encoding methods in a one-to-one manner. This resulted in a total of 144 BMs, each of which produced prediction probability scores (PBS) in 5-fold cross-validation (CV) trial. To develop the final meta-prediction model, we used the LR model as a meta-learning method. The meta-predictor generates the final probability scores by analyzing the PBS of BMs. The meta-predictor model is as follows:

$$\log\left(\frac{P}{1-P}\right) = \beta_0 + \beta_1 X_1 + \beta_2 X_2 + \ldots + \beta_n X_n \tag{20}$$

where $\beta_o$ and $\beta_i$ are the regression coefficients, $X_i$ is the probability scores generated by the $i$-th BMs and $n = 144$ is the total number of the BMs. The meta-predictor generates probability scores between 0 and 1, with a score of 0.5 or higher indicating a 2-OM site and a score below 0.5 indicating a non-2-OM site.

### Performance evaluation

To assess the performance of the prediction models, we used seven statistical measures: sensitivity (SEN), specificity (SPE), precision (PRE), ACC, Matthew's correlation coefficient (MCC), AUC, and area under the precision-recall curve (AUPRC). Most compatible measures were defined as

$$SEN = \frac{TP}{TP + FN} \tag{21}$$

$$SPE = \frac{TN}{TN + FP} \tag{22}$$

$$PRE = \frac{TP}{TP + FP} \tag{23}$$

$$ACC = \frac{TP + TN}{TP + FN + FP + TN} \tag{24}$$

$$MCC = \frac{TP \times TN - FP \times FN}{\sqrt{(TN + FN) \times (TP + FP) \times (TN + FP) \times (TP + FN)}} \tag{25}$$

where TP is the number of true positives, FP is the number of false positives, TN is the number of true negatives, and FN is the number of false negatives. The receiver operating characteristic curve (ROC) is also utilized to visualize the prediction performance of the model according to 5-fold CV.

## Results

### Nucleotide preference analysis

The nucleotide patterns surrounding the central position in sequence data play a pivotal role in classification tasks. We used the kpLogo command-based tool [80] to perform sequence logo analysis to examine the nucleotide distribution among the 2-OM (positive) and non-2-OM (negative) samples. Fig 2 shows the probability logo (pLogo) and *k*-mer probability logo (*k*pLogo) for k-mer of 1 to 4, highlighting the most significantly enriched or depleted base pairs and sequence motifs, respectively. The pLogo result suggests that the base pairs A, G, A, and U were highly enriched at positions 22, 23, 24, and 25, respectively, for 2-OM sites. Conversely, the base pairs G, A, G, and G are depleted at the same positions 22, 23, 24, and 25, respectively, for non-2-OM sites (Fig 2A). The *k*pLogo results show that the 2-OM samples presented the most significant motifs of ACAG, AGAU, GAUC, AUCG, UCGG, CGG, GGAA, and GAAG, which occurred at positions 20, 22, 23, 24, 25, 26, 27, and 28, respectively (Fig 2B). The depletion motifs were the same as the pLogo findings. Overall, the enrichment and depletion results mostly occurred on the upstream side of positive and negative samples. These results suggested that the distinct nucleotide preferences and unique motifs made it possible to differentiate RNA 2-OM modification/non-modification sites.

### Baseline models: Construction and evaluation

ML classifiers effectively leverage feature information from the training dataset to categorize the class. However, the performance of each classifier can vary depending on the feature encodings used, as each encoding has unique characteristics [69, 77]. In this study, we employed eighteen different feature encoding algorithms (Binary, DNC, TNC, RCKmer, ENAC, CKSNAP, ANF, NCP, EIIP, PseEIIP, Kmer, BPB, CTD, NCP-ND, NPS, NPPS, PseKNC, W2V), and assessed their inherent patterns between 2-OM and non-2-OM sites using eight ML classifiers, including four tree-based classifiers (RF, LGBM, XGB, CBC), two decision boundary-based classifier (SVM, LR), and others (NB, KNN). Notably, we trained each model via a 5-fold CV and subsequently evaluated the optimal model using an independent dataset. Firstly, we conducted the performance analysis by using generic datasets. Detailed performance of the 144 baseline models via CV and their transferability on the

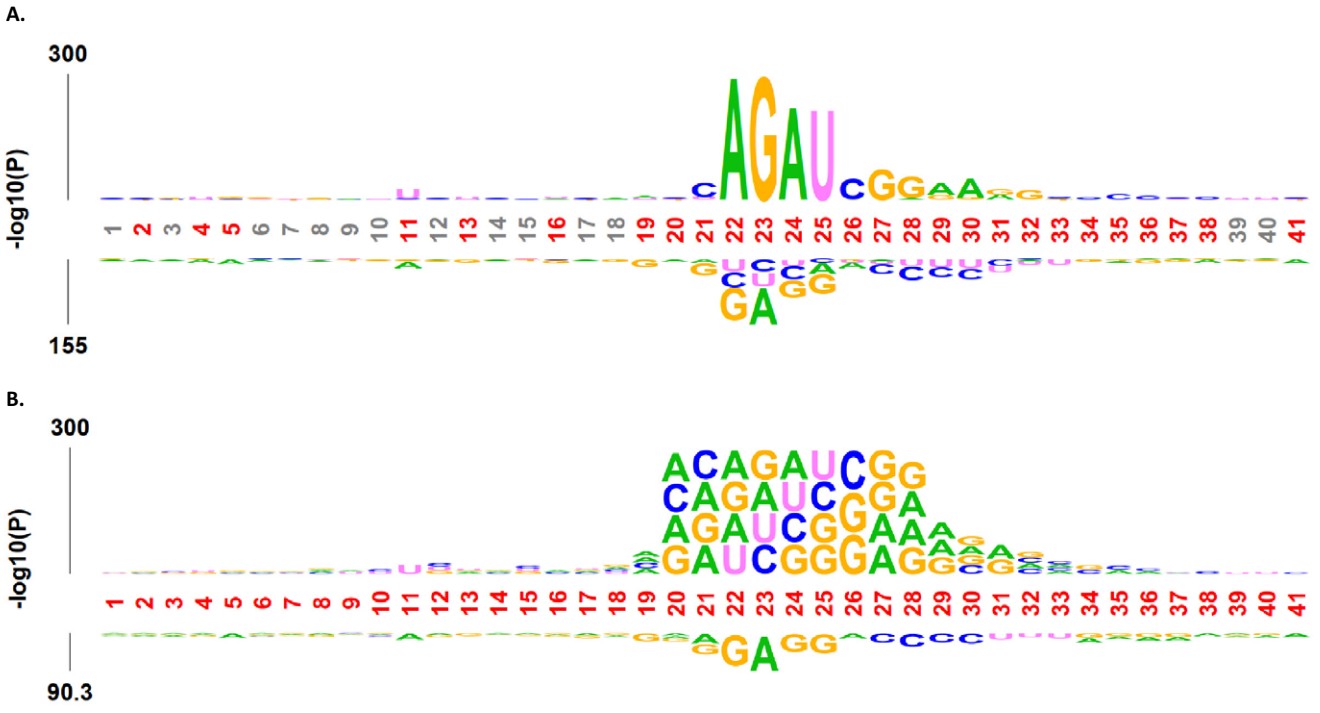

**Fig 2. Nucleotide preference pattern analysis of 2-OM RNA modification sites to understand the effect of base pairs and motifs of sequence data.** (A) pLogo, and (B) kpLogo.

independent dataset are shown in S1 and S2 Figs in S1 File and S2 Table. We observed that the overall performance trend of each classifier with 18 different encodings was similar between the training and independent datasets. We observed that W2V, NPPS, and Binary were the top encoding algorithms. The LGBM classifiers achieved very high performance with an average MCC of 0.604 across 18 encodings, SVM and LR provided MCCs of 0.585 and 0.535. CBC, RF, NB, KNN and XGB MCCs of 0.580, 0.516, 0.491, 0.467 and 0.505, respectively. These results indicate that all eight ML algorithms worked well on the given task.

To assess the performance of the baseline models, we evaluated three global statistical measures (MCC, ACC, AUC) on the training and independent datasets for generic cases (Fig 3). The performance of the baseline models in each ML showed a similar trend with respect to encodings. RFs with DNC, TNC, PseEIIP, Binary, NCP-ND, BPB, NPPS, PseKNC, and W2V encodings surpassed MCC of 0.550, ACC of 0.750, and AUC of 0.820 for both the training and independent datasets. The SVM with W2V exceeded the performance on MCC of 0.710, ACC of 0.850, and AUC of 0.930. In contrast, the XGB with CKSNAP, NPS, NPPS, and W2V attained performance exceeding MCC of 0.550, ACC of 0.770, and AUC of 0.840 during training and testing. Also, the W2V encoding revealed significant prediction performance with CBC and LR where overcoming the MCC of 0.650, ACC of 0.820, and AUC of 0.910. Importantly, the LGBM with Binary, NCP, EIIP, BPB, NPPS, and W2V showed high performance compared to other baseline models, where they produced a surpassed MCC of 0.640, ACC of 0.820, and AUC of 0.900 on training and independent datasets. Among the 144 baseline models, the best single model was LGBM_W2V. The common thought is that an optimal single-feature classifier is considered as the final prediction model. Generally, a single-feature classifier varies the performance, depending on the size of the datasets [81, 82]. For example, they sometimes fail to perform well on large-scale datasets due to the dimensionality of the data

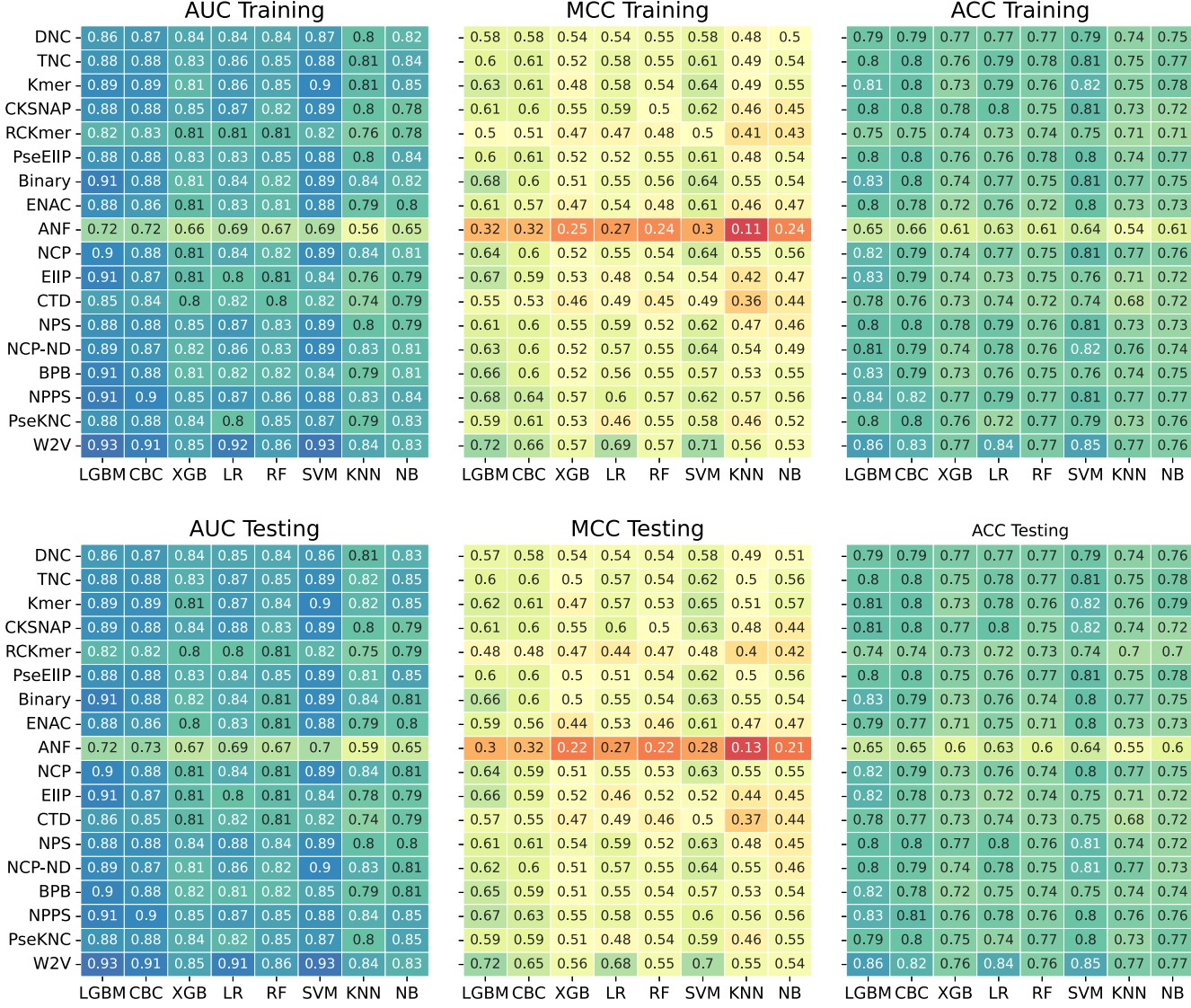

**Fig 3. Performance analysis of the baseline classifiers.** The classifiers were generated through eight different MLs with eighteen single-feature encoding methods. The MCC, ACC, and AUC are presented on the training (A, B, C) and independent (D, E, F) datasets.

points. Therefore, a meta-learning ensemble strategy was considered to stack the baseline or single-feature models for enhanced prediction.

Next, we trained the meta-classifiers implementing multiple baseline models on the specific (Am, Cm, Gm, Um) datasets and evaluated their performance on the test dataset. Three statistical measures (MCC, ACC, AUC) were illustrated for respective specific nucleotide modifications in S3-S6 Figs in S1 File. The LGBM_W2V and SVM_W2V were regarded as the best baseline models for all nucleotide-specific datasets.

## Impact of meta-learning on the development of Meta-2OM

The stacking approach represents an advanced ensemble method that enhances prediction performance by amalgamating the strengths of multiple models [45, 64, 83–87]. Unlike conventional ensemble approaches that primarily use averaging and voting, stacking employs a

**Table 3. Ranking of the baseline models in the descending order of the AUC values.**

| Rank | Baseline model | Rank | Baseline model | Rank | Baseline model | Rank | Baseline model | Rank | Baseline model |
|---|---|---|---|---|---|---|---|---|---|
| 1 | LGBM_W2V | 31 | CBC_PseEIIP | 61 | NB_Kmer | 91 | LGBM_RCKmer | 121 | RF_CTD |
| 2 | SVM_W2V | 32 | CBC_TNC | 62 | RF_Kmer | 92 | SVM_CTD | 122 | KN_NPS |
| 3 | LR_W2V | 33 | LGBM_PseEIIP | 63 | LR_DNC | 93 | RF_BPB | 123 | LR_EIIP |
| 4 | LGBM_NPPS | 34 | LGBM_TNC | 64 | KN_W2V | 94 | NB_DNC | 124 | LR_PseKNC |
| 5 | CBC_W2V | 35 | CBC_NCP | 65 | CBC_CTD | 95 | LR_BPB | 125 | KN_CKSNAP |
| 6 | LGBM_Binary | 36 | CBC_PseKNC | 66 | NB_NPPS | 96 | NB_Binary | 126 | KN_PseKNC |
| 7 | LGBM_EIIP | 37 | LGBM_PseKNC | 67 | RF_DNC | 97 | XGB_NCP-ND | 127 | NB_EIIP |
| 8 | LGBM_BPB | 38 | CBC_NCP-ND | 68 | XGB_DNC | 98 | RF_Binary | 128 | NB_NPS |
| 9 | CBC_NPPS | 39 | LR_CKSNAP | 69 | NB_PseEIIP | 99 | RF_NCP | 129 | KN_ENAC |
| 10 | SVM_Kmer | 40 | CBC_EIIP | 70 | NB_TNC | 100 | LR_CTD | 130 | KN_BPB |
| 11 | LGBM_NCP | 41 | LR_NPPS | 71 | LR_Binary | 101 | XGB_Kmer | 131 | NB_CTD |
| 12 | SVM_NCP-ND | 42 | LR_NPS | 72 | LR_NCP | 102 | NB_NCP-ND | 132 | NB_CKSNAP |
| 13 | LGBM_Kmer | 43 | CBC_DNC | 73 | XGB_PseKNC | 103 | NB_NCP | 133 | NB_RCKmer |
| 14 | LGBM_NCP-ND | 44 | SVM_DNC | 74 | SVM_EIIP | 104 | XGB_Binary | 134 | KN_EIIP |
| 15 | SVM_NCP | 45 | SVM_PseKNC | 75 | SVM_BPB | 105 | NB_BPB | 135 | KN_RCKmer |
| 16 | SVM_Binary | 46 | CBC_ENAC | 76 | KN_Binary | 106 | XGB_BPB | 136 | KN_CTD |
| 17 | SVM_CKSNAP | 47 | LR_Kmer | 77 | KN_NCP | 107 | RF_ENAC | 137 | CBC_ANF |
| 18 | SVM_NPS | 48 | LGBM_DNC | 78 | KN_NPPS | 108 | KN_Kmer | 138 | LGBM_ANF |
| 19 | CBC_Kmer | 49 | LR_TNC | 79 | RF_NPS | 109 | RF_EIIP | 139 | SVM_ANF |
| 20 | SVM_NPPS | 50 | RF_W2V | 80 | RF_NCP-ND | 110 | XGB_NCP | 140 | LR_ANF |
| 21 | LGBM_ENAC | 51 | LR_NCP-ND | 81 | XGB_TNC | 111 | RF_RCKmer | 141 | RF_ANF |
| 22 | SVM_TNC | 52 | RF_NPPS | 82 | XGB_PseEIIP | 112 | KN_TNC | 142 | XGB_ANF |
| 23 | SVM_PseEIIP | 53 | LGBM_CTD | 83 | LR_PseEIIP | 113 | XGB_EIIP | 143 | NB_ANF |
| 24 | CBC_Binary | 54 | XGB_NPPS | 84 | NB_PseKNC | 114 | XGB_RCKmer | 144 | KN_ANF |
| 25 | SVM_ENAC | 55 | XGB_W2V | 85 | LR_ENAC | 115 | XGB_ENAC | | |
| 26 | CBC_NPS | 56 | RF_TNC | 86 | KN_NCP-ND | 116 | LR_RCKmer | | |
| 27 | LGBM_CKSNAP | 57 | RF_PseEIIP | 87 | NB_W2V | 117 | NB_ENAC | | |
| 28 | LGBM_NPS | 58 | RF_PseKNC | 88 | CBC_RCKmer | 118 | KN_PseEIIP | | |
| 29 | CBC_CKSNAP | 59 | XGB_CKSNAP | 89 | RF_CKSNAP | 119 | XGB_CTD | | |
| 30 | CBC_BPB | 60 | XGB_NPS | 90 | SVM_RCKmer | 120 | KN_DNC | | |

meta-model to adeptly combine the forecasts from its base models. This meta-model is trained using the base models' outputs, optimizing their inputs to generate a refined, often superior, final prediction [88, 89]. The notable ACC of stacking is attributed to its proficiency in maximizing the positives of different classifiers while offsetting their limitations. In our research, we harnessed stacking to amalgamate predictions from eight distinct models. Through a meta-learner, which identified patterns among individual predictions, we drew a diverse set of features derived from RNA sequences and leveraged the unique strengths of each classifier to elevate the overall model efficiency. Firstly, we ranked the 144 baseline models by their AUC (Table 3). We then used these rankings to stack the probability scores of the generic (Nm) baseline models using LR. Specifically, we developed 20 meta-classifiers, leveraging the top 1, 2, 4, 8, 12, 16, 20, 30, 40, 50, 60, 70, 80, 90, 100, 110, 120, 130, 140, and all 144 baseline models. Fig 4 shows that the performance of these meta-classifiers progressively improves as the number of probabilistic features (baseline models) increases, both in the training and independent datasets. Our analysis indicating that employing all baseline models, our meta-classifiers achieved the highest AUC on the training dataset as well as the independent dataset.

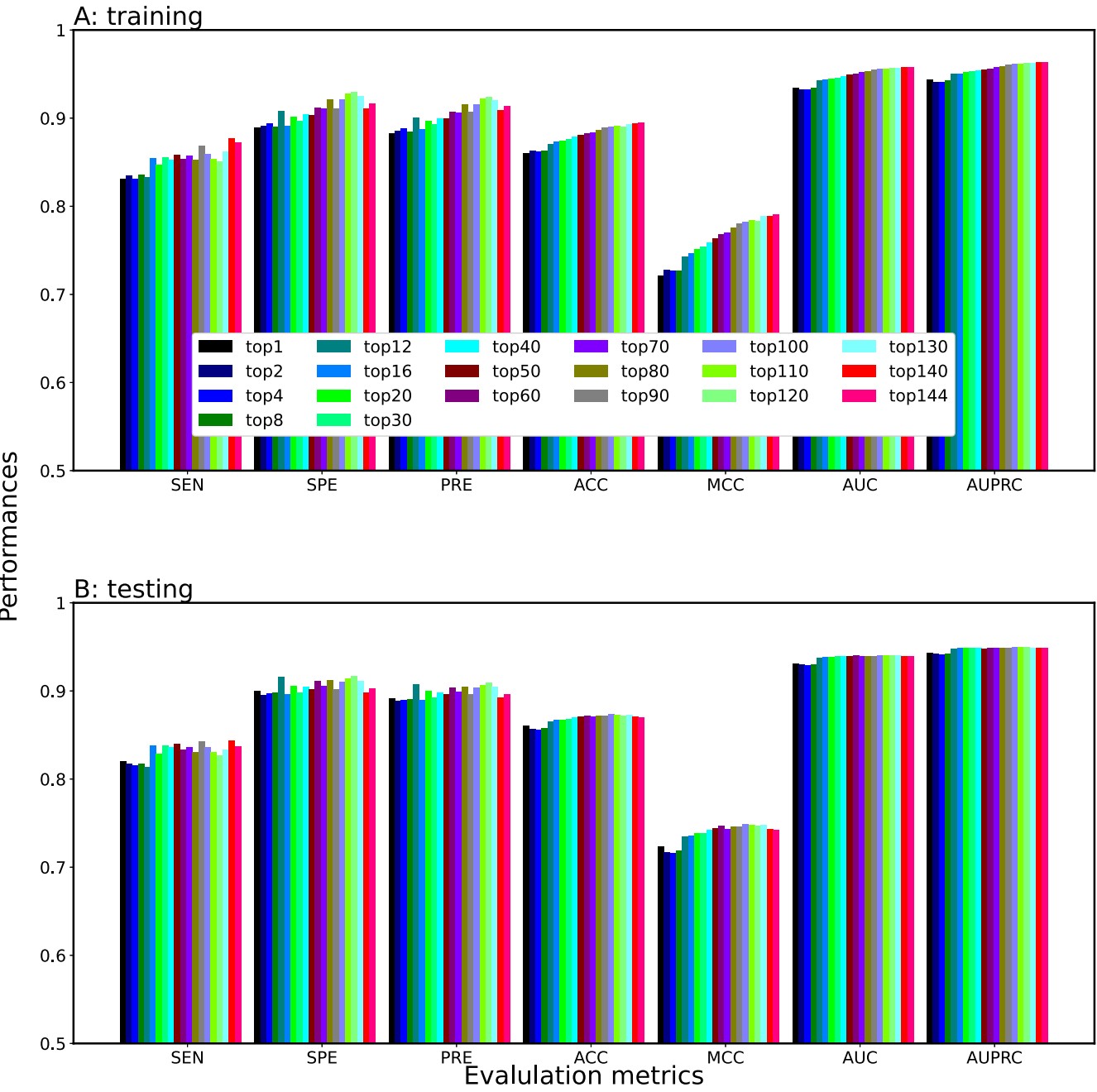

**Fig 4. Effect of the employed feature sets on meta-classifiers.** The 20 meta-classifiers were generated. The performance was compared based on (A) training and (B) independent datasets.

To evaluate the effectiveness of various meta-classifiers, we employed six additional ML methods (LGBM, XGB, RF, SVM, KNN, NB) as a meta-classifier to stack baseline models. We then assessed and compared their performance metrics, as provided in Table 4. The results showed that the LR-based meta-classifier, which aggregated predictions from 144 baseline models, surpassed all other ML methods-based stacking models in terms of performance on independent test datasets. Conversely, the other meta-classifiers exhibited signs of overfitting,

**Table 4. Performance comparison among seven meta-classifiers incorporating the 144 baseline models on the training and test datasets.**

| Meta-classifier | Training | | | | | Test | | | | |
|---|---|---|---|---|---|---|---|---|---|---|
| | SEN | SPE | ACC | MCC | AUC | SEN | SPE | ACC | MCC | AUC |
| LR | 0.871 | 0.918 | 0.894 | 0.791 | 0.958 | **0.836** | 0.904 | **0.870** | **0.743** | **0.940** |
| LGBM | 0.960 | 0.975 | 0.968 | 0.936 | **0.996** | 0.792 | 0.893 | 0.843 | 0.692 | 0.923 |
| XGB | 0.962 | **0.980** | 0.971 | 0.942 | 0.975 | 0.805 | 0.910 | 0.857 | 0.719 | 0.873 |
| RF | 0.962 | 0.977 | 0.970 | 0.940 | 0.994 | 0.803 | 0.898 | 0.850 | 0.704 | 0.922 |
| SVM | **0.967** | 0.978 | **0.973** | 0.945 | **0.997** | 0.820 | 0.903 | 0.862 | 0.727 | 0.933 |
| KNN | 0.964 | **0.981** | **0.973** | **0.946** | 0.976 | 0.814 | **0.915** | 0.864 | 0.733 | 0.879 |
| NB | 0.964 | 0.979 | 0.971 | 0.943 | 0.972 | 0.816 | 0.905 | 0.860 | 0.723 | 0.862 |

Bold indicates the highest score for the corresponding statistical measure

demonstrated by superior performance on the training datasets compared to the test dataset. Therefore, we selected the LR-based meta-model incorporating all baseline models, as our final predictor, and named Meta-2OM. Meta-2OM achieved the best performance with SEN of 0.871, SPE of 0.918, ACC of 0.894, MCC of 0.791, PRE of 0.915, and AUC of 0.958 on the training dataset. On the independent datasets, it achieved a SEN of 0.836, SPE of 0.904, ACC of 0.870, MCC of 0.743, PRE of 0.898, and AUC of 0.940 (Table 4). Compared to the best-performing baseline model, LGBM_W2V, Meta-2OM demonstrated improvements of 9.5% in MCC, 4.1% in ACC, 2.6% in AUC on the training dataset, alongside increases of 2.8% in MCC, 1.3% in ACC, and 1.0% in AUC on the independent dataset, thus substantiating the advanced efficacy of our proposed method.

## Comparison of performance based on generic and nucleotide-specific modifications

The 2-OM (Nm) datasets contained four distinct nucleotides (Am, Cm, Gm, Um) modifications. To ascertain the efficacy of Meta-2OM for each type of nucleotide modification, we employed LR-based meta-classifiers, previously demonstrated as the most effective in the generic model development, across all specific nucleotide modifications. Each nucleotide modification dataset was divided into the same 20 groups as in the generic case and corresponding the stacking models were developed. The performances of these models are shown in S7-S10 Figs in S1 File, with the meta-classifier aggregating all 144 baseline models achieved the highest AUC score across all nucleotide-specific cases. Moreover, the capability of the generically trained model (Meta-2OM) to distinguish between positive and negative modification sites was compared against that of the nucleotide-specific model using the independent dataset, as shown in Fig 5 and S3 Table. Remarkably, the generic model (Meta-2OM) more accurately classified the positive and negative sites than nucleotide-specific models. This suggests that Meta-2OM effectively identifies both collective (Nm) and individual nucleotide modifications (Am, Cm, Gm, Um) across comprehensive transcriptome-wide datasets, highlighting its robust applicability and superior predictive performance.

## Comparison of Meta-2OM with existing methods

To demonstrate the superiority of Meta-2OM, we conducted a comparative analysis against existing state-of-the-art methods [32–39, 42]. Several methods were excluded from this comparison due to the use of different datasets, inactive web services, or the absence of standalone software packages. Additionally, we did not include methods such as NmRF [32], the approach

## 7-measures performance profiles on independent set

**Fig 5. Comparison of prediction performance between Meta-2OM generic model and nucleotide-specific model under four different specific nucleotide modifications independent datasets.**

by Huang et al. [34], iRNA-PseKNC (2methyl) [35], iRNA-2OM [37], and Chen et al.'s method [39], as these were trained exclusively on datasets featuring adenine nucleotides (Am) at the central positions, unlike Meta-2OM which can predict Nm modification sites for all four nucleotides.

**Table 5. Comparison of Meta-2OM with state-of-the-art methods.**

| Method | Am | | | | | Cm | | | | |
|---|---|---|---|---|---|---|---|---|---|---|
| | SEN | PRE | ACC | MCC | AUC | SEN | PRE | ACC | MCC | AUC |
| i2OM | **0.908** | 0.135 | 0.461 | 0.192 | 0.862 | 0.774 | 0.425 | 0.884 | 0.518 | 0.915 |
| H2Opred | 0.863 | 0.906 | 0.887 | 0.774 | 0.943 | **0.849** | 0.912 | 0.883 | 0.769 | 0.946 |
| Meta-2OM | 0.856 | **0.918** | **0.889** | **0.781** | **0.946** | 0.843 | **0.936** | **0.892** | **0.789** | **0.953** |
| Method | Gm | | | | | Um | | | | |
| | SEN | PRE | ACC | MCC | AUC | SEN | PRE | ACC | MCC | AUC |
| i2OM | 0.830 | 0.431 | 0.885 | 0.545 | 0.934 | 0.597 | 0.227 | 0.779 | 0.266 | 0.774 |
| H2Opred | **0.853** | **0.906** | **0.882** | **0.765** | 0.946 | **0.831** | 0.809 | 0.818 | 0.636 | 0.898 |
| Meta-2OM | 0.850 | 0.903 | 0.879 | 0.759 | **0.946** | 0.796 | **0.837** | **0.820** | **0.642** | **0.907** |
| Method | Average value | | | | | | | | | |
| | SEN | PRE | ACC | MCC | AUC | | | | | |
| i2OM | 0.777 | 0.305 | 0.752 | 0.380 | 0.871 | | | | | |
| H2Opred | **0.849** | 0.883 | 0.867 | 0.736 | 0.933 | | | | | |
| Meta-2OM | 0.836 | **0.899** | **0.870** | **0.743** | **0.938** | | | | | |

Bold value indicates the highest score for the corresponding statistical measure; Average value indicates Mean(Am, Cm, Gm, Um) under respective measures

Finally, we selected the i2OM method [41] and H2Opred [42], trained on larger datasets, that can predict both Nm modification sites and specific nucleotide-based sites. The results demonstrated that Meta-2OM slightly better than H2Opred and significantly better than i2OM in identifying Nm modification sites as well as individual types of nucleotide modification sites (Table 5 and Fig 6). Upon assessing the performance across individual nucleotide datasets, Meta-2OM demonstrated marginal enhancements in the Am, Cm, and Um datasets when compared to H2Opred. However, in the case of Gm, H2Opred exhibited superior performance relative to Meta-2OM. Additionally, Meta-2OM significantly surpassed the performance of i2OM across these assessments.

Furthermore, Meta-2OM also outperformed single ML-based methods, such as RF-based NmRF [32], RF-based NmSEER V2.0 [36], SVM-based Huang et al. [34], and CNN-based Deep-2'-O-Me [38], in predicting Nm sites on the independent dataset, achieving an AUC of 0.940. Overall, the meta-learning approach implemented in the Meta-2OM method is responsible for its improved performance compared to the existing predictors.

## Probabilistic-feature analysis

Meta-2OM exhibited superior performance compared to the single-feature models on both the training and independent datasets. To delve deeper into how the model works, we conducted SHapley Additive exPlanation (SHAP) analysis on the probabilistic features generated by the 144 baseline models. The most influential 20 are shown in Fig 7. The importance ranking of the baseline models revealed that classifiers such as LGBM, CBC, LR, and SVM methods contributing more to the prediction than RF, NB, KNN and XGB. Notably, the W2V based LGBM_W2V, LR_W2V, SVM_W2V, and CBC_W2V among the top 20 made significant contribution, while the remaining models played a complementary role on Meta-2OM prediction. To further demonstrate the Meta-2OM prediction capability, we applied the t-distributed stochastic neighbor embedding (t-SNE; scikit-learn v.1.0.2) to compute the 2D feature based on the 144D probabilistic features, as shown in Fig 8. The results demonstrated a clear separation between positive and negative samples, indicating that the Meta-2OM effectively discerns the underlying patterns distinguishing 2-OM from non-2-OM modification sequences.

## The comparison of SOTA methods on independent set

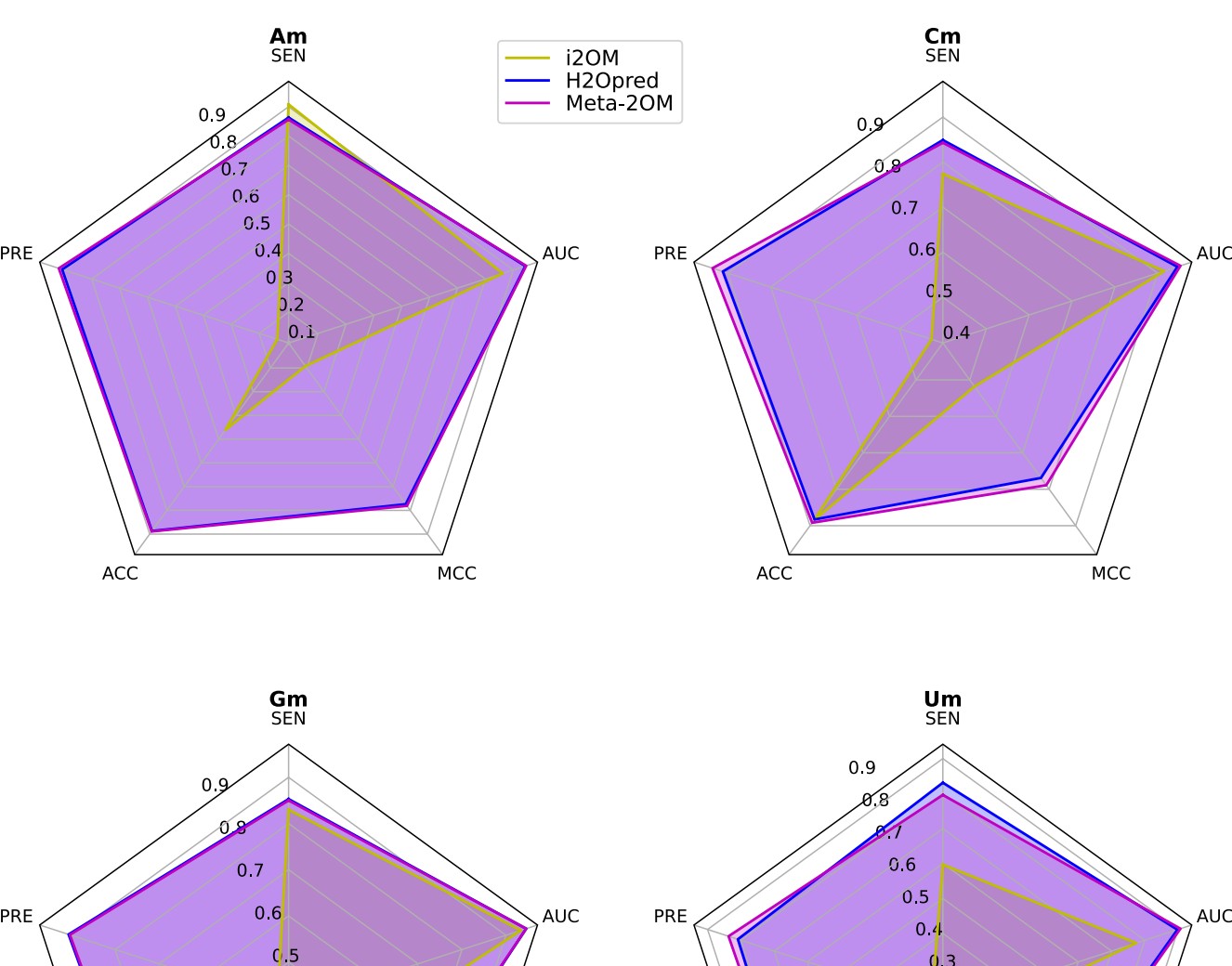

**Fig 6. Parallel comparison of the Meta-2OM predictors with state-of-the-art methods based on the nucleotide-specific independent datasets.**

## Web server implementation

We developed a user-friendly and freely accessible web server for Meta-2OM, which is available at http://kurata35.bio.kyutech.ac.jp/Meta-2OM/. The web server is implemented using Flask (2.2.2) in the Python program and Apache (2.4.52). Users can easily carry out the prediction by submitting FASTA sequences of 41 bp in length. The prediction results can be

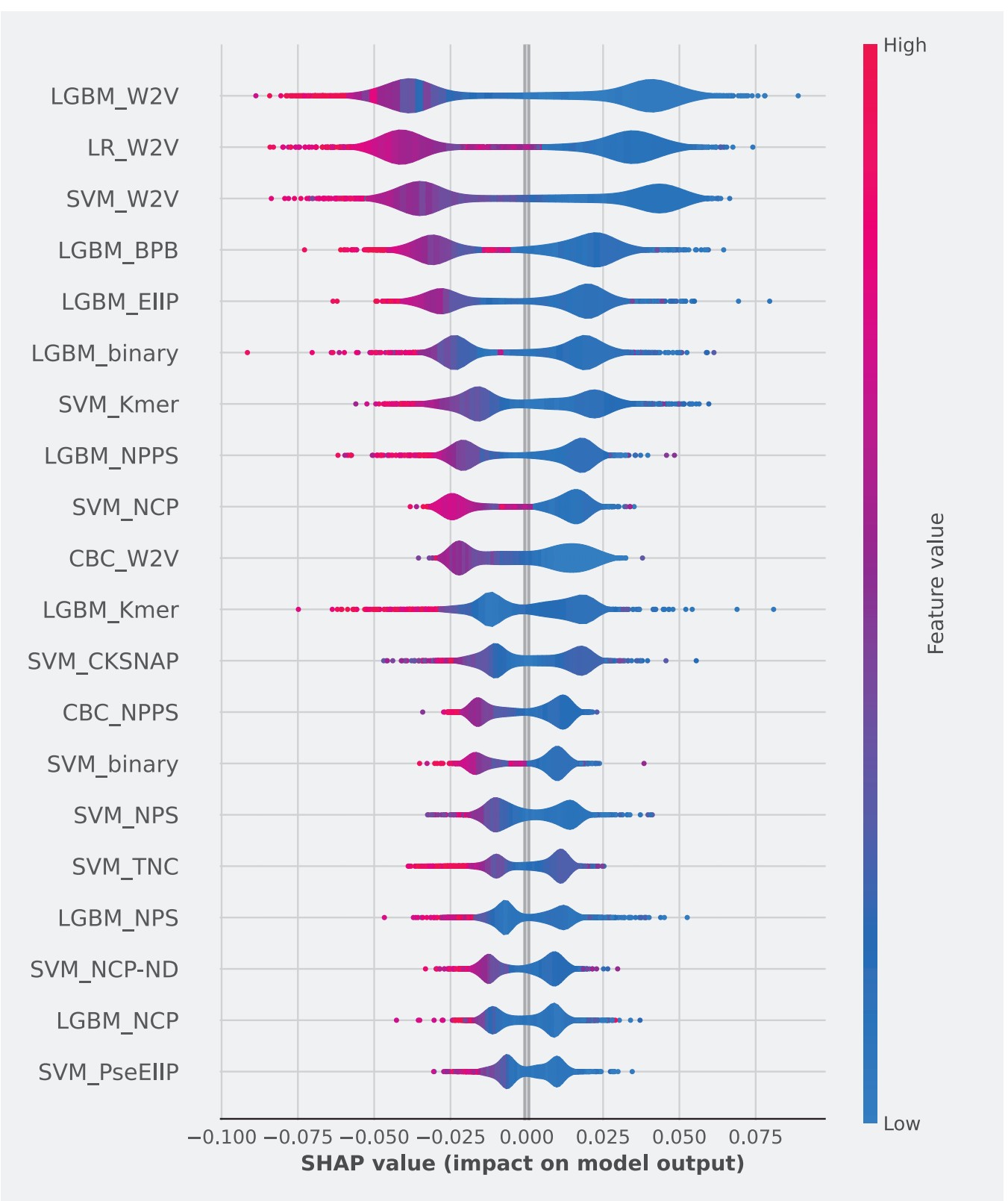

**Fig 7. SHAP plot analysis using the probability features generated by 144 baseline models considered in Meta-2OM.** The top 20 were illustrated. The position on the y-axis is determined by the feature and that on the x-axis by the SHAP value. The color (red to blue) represents the value of the feature from low to high. The SHAP's positive and negative values mean that they are associated with a higher and lower prediction, respectively. The features are ordered according to their importance.

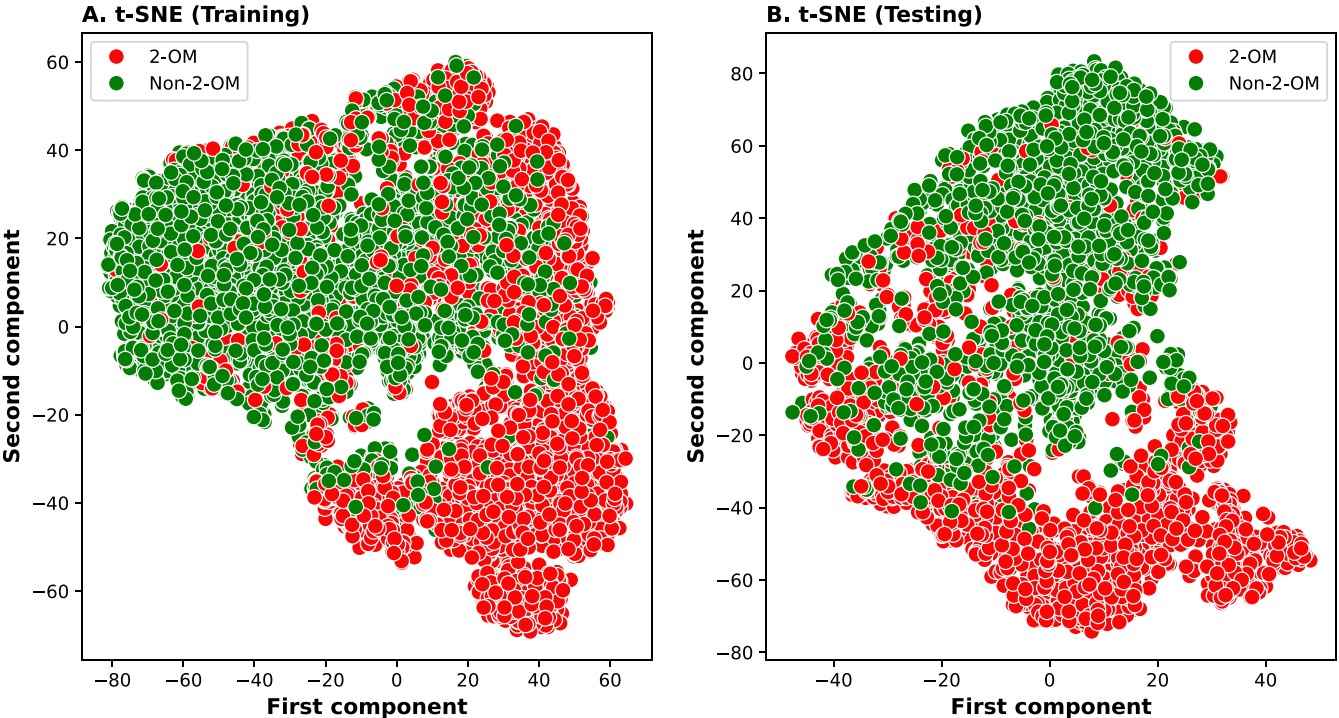

**Fig 8. Analysis of the probability features generated by 144 baseline models for the negative and positive samples by t-distributed stochastic neighbor embedding (t-SNE).** It represents the discriminative distribution of prediction probabilities. These results are shown for both (A) training and (B) independent datasets.

downloaded as an Excel file (.csv) with three columns: predicted labels, predicted probability scores, and sample sequences. The threshold value for predicting Nm modification site is 0.5. Detailed user guidelines are available on the website's help page.

## Discussion and conclusion

2-OM modification is a prevalent and crucial modification in RNA, necessary for various biological and functional mechanisms. Accurate identification of 2-OM sites is essential to further understanding its roles. The biological experimental methods that identify 2-OM sites are time-consuming, labor-intensive, and expensive [20–22, 24]. To complement such experimental methods, we have developed a computational model to predict 2-OM sites, using benchmark datasets covering all four types of nucleotide modification (Am, Cm, Um, Gm) in the human chromosomes. Specifically, a total of 12,182 positive and negative samples were employed to build the model, where each sample consisted of 41 nucleotide base pairs.

In this work, we developed Meta-2OM, a meta-learning-based ML approach for accurately identifying 2-OM RNA modification sites using sequence information. We first assessed the prediction performance of eight widely established ML classifiers, where each classifier trained with 18 types of RNA sequence features. Subsequently, we built a meta-model that integrates the predictions of the baseline models to enhance the overall prediction accuracy. Meta-2OM demonstrated superior performance compared to the existing predictors when tested on an independent dataset. The efficiency of Meta-2OM is attributed to the following factors: (i) utilization of 18 different feature encoding algorithms that capture various intrinsic capabilities to discriminate between 2-OM and non-2-OM sites; and (ii) construction of meta-model incorporating 144 baseline models. Therefore, we believe that the current approach can be

extended to other sequence-based function prediction problems, including enhancer prediction, peptide therapeutic function prediction, replication original sites prediction, and post-translational modification sites prediction [90–93]. While Meta-2OM achieves an ACC exceeding 87% on both training and independent datasets, we expect that incorporating position-specific information or natural language processing-based features with variants DL algorithms may enhance the predictive performance. Currently, the proposed Meta-2OM predictors do not consider RNA tertiary structures, which are crucial for accurate RNA site prediction. By integrating information about RNA 3D structures into our model, we could significantly increase its predictive accuracy. We plan to explore these potential improvements in our future studies.

## Supporting information

**S1 Table. Hyperparameter tuning for each machine-learning method.**
(PDF)

**S2 Table. Sequential performance of 144 baseline classifiers through the training datasets.**
(PDF)

**S3 Table. The individual performance of the nucleotide-specific model and a comparison with the generic model under training and test datasets.**
(PDF)

**S1 File. The several performance analysis representations of 8 ML techniques with 18 different encoding models.** Also, the stacking performance representation is based on nucleotide-specific datasets.
(PDF)

## Author Contributions

**Conceptualization:** Md. Harun-Or-Roshid, Hiroyuki Kurata.

**Data curation:** Md. Harun-Or-Roshid.

**Formal analysis:** Md. Harun-Or-Roshid, Hiroyuki Kurata.

**Funding acquisition:** Balachandran Manavalan, Hiroyuki Kurata.

**Investigation:** Md. Harun-Or-Roshid.

**Methodology:** Md. Harun-Or-Roshid, Hiroyuki Kurata.

**Project administration:** Balachandran Manavalan, Hiroyuki Kurata.

**Resources:** Md. Harun-Or-Roshid, Hiroyuki Kurata.

**Software:** Md. Harun-Or-Roshid.

**Supervision:** Balachandran Manavalan, Hiroyuki Kurata.

**Validation:** Md. Harun-Or-Roshid.

**Visualization:** Md. Harun-Or-Roshid, Nhat Truong Pham.

**Writing – original draft:** Md. Harun-Or-Roshid.

**Writing – review & editing:** Nhat Truong Pham, Balachandran Manavalan, Hiroyuki Kurata.

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
