## [Decision Letter · Decision Letter 0]

14 Feb 2024

PONE-D-24-02313Meta2-OM: A multi-classifier meta-model for the accurate prediction of RNA 2'-O-methylation sites in human RNAPLOS ONE

Dear Dr. Kurata,

Thank you for submitting your manuscript to PLOS ONE. After careful consideration, we feel that it has merit but does not fully meet PLOS ONE’s publication criteria as it currently stands. Therefore, we invite you to submit a revised version of the manuscript that addresses the points raised during the review process.

ACADEMIC EDITOR: Major Revision/>==============================

We look forward to receiving your revised manuscript.

Kind regards,

Shahid Akbar, PhD

Academic Editor

PLOS ONE

Journal Requirements:

“This work was supported by a Grant-in-Aid for Scientific Research (B) (22H03688) from the Japan Society for the Promotion of Science (JSPS), Japan. It was also supported by the National Research Foundation of Korea (NRF) funded by the Ministry of Science and ICT (2021R1A2C1014338), Republic of Korea.”

Reviewers' comments:

Reviewer's Responses to Questions

Comments to the Author

1. Is the manuscript technically sound, and do the data support the conclusions?

Reviewer #1: Yes

Reviewer #2: Yes

2. Has the statistical analysis been performed appropriately and rigorously? 

Reviewer #1: Yes

Reviewer #2: Yes

3. Have the authors made all data underlying the findings in their manuscript fully available?

Reviewer #1: Yes

Reviewer #2: Yes

4. Is the manuscript presented in an intelligible fashion and written in standard English?

Reviewer #1: Yes

Reviewer #2: Yes

5. Review Comments to the Author

Reviewer #1: Roshid et al. have created the Meta2OM predictor, employing a meta-learning-based stacking approach. They also conducted a comparative analysis with existing methods, demonstrating superior performance. Additionally, they have made a significant contribution to the research community by providing both a web server and a standalone program. However, I have the following suggestions for further improvement.

1.Improve the overall flow and structure of the manuscript. Ensure all figures are high-quality and meet journal publication standards.

2.Investigate the potential benefits of using NLP-based embeddings and additional feature descriptors beyond those employed in previous studies. Experiment with various feature combinations to optimize performance.

3.Explore diverse classifier architectures beyond those used in previous works. Consider incorporating new algorithms and comparing their effectiveness.

4.Provide details on the parameter search ranges used during model development. Include this information in a dedicated README file for transparency and reproducibility.

5.Utilize all relevant information from baseline models, including probabilistic scores, class labels, and their combinations. Explore different ways of incorporating this information to enhance prediction performance.

6.Use acronyms consistently throughout the manuscript (e.g., always use ML or Machine Learning, ACC or Accuracy). Define acronyms upon first use and avoid unnecessary acronyms.

7.Explain the reasoning behind using a CD-HIT threshold of 0.8. Provide justification for this specific value and its impact on the results.

8.For the user concerns to provide a clear background of training models, authors are advised to cite the recent predictors such as AIPs-SnTCN , iACP-GAEnsC, iACP-GAEnsC, pAtbp-EnC, and cACP-DeepGram

9.Change the title to "Meta-2OM" for consistency and clarity.

Reviewer #2: This study presents promising Meta2-OM predictors for identifying 2OM sites using meta-learning techniques. The careful dataset construction and exploration of diverse ML classifiers and encodings suggest a well-considered approach. Given the growing interest in RNA post-transcriptional modification sites within bioinformatics, this work holds significant potential. To further strengthen its impact, I have the following suggestions:

1.      The authors should investigate both nucleotide-specific and generic models. Compare their advantages and disadvantages, highlighting the strengths of each approach and justifying the choice of using a generic dataset for this study.

2.      Clearly explain how the current dataset differs from the one used in i2OM. This will provide context and allow for a more meaningful comparison of the two methods.

3.      Include H2Opred in the comparison table alongside other existing methods. Analyze its performance relative to your proposed tool to provide a more comprehensive evaluation.

4.      Comparison of Meta2-OM with existing methods. I have noticed they have used just only one method. It should be revised.

5.      Revise Table 5 to ensure a fair comparison with i2OM. If relevant, consider incorporating nucleotide-specific models from both methods as suggested in point 1.

6.In the machine learning section, the authors should incorporate the following predictors such as iMethyl-STTNC,DP-BINDER, and AFP-CMBPred to provide clear understanding of the model.

7.      Analyze the optimal sequence length for prediction accuracy. Construct baseline models with varying sequence lengths (e.g., 31-201 bp) and compare their performance to identify the most effective length.

6. PLOS authors have the option to publish the peer review history of their article (what does this mean?). If published, this will include your full peer review and any attached files.

Do you want your identity to be public for this peer review? For information about this choice, including consent withdrawal, please see our Privacy Policy.

Reviewer #1: No

Reviewer #2: No

---

## [Author Response · Author response to Decision Letter 0]

21 May 2024

Dear Academic Editor

PLOS ONE

Reviewer #1

Roshid et al. have created the Meta2OM predictor, employing a meta-learning-based stacking approach. They also conducted a comparative analysis with existing methods, demonstrating superior performance. Additionally, they have made a significant contribution to the research community by providing both a web server and a standalone program. However, I have the following suggestions for further improvement.

1. Improve the overall flow and structure of the manuscript. Ensure all figures are high-quality and meet journal publication standards. 

Author’s response: As per your suggestion, we revised the flow and structure of the manuscript. We ensured that the figure files met journal publication standards for quality. Additionally, we confirmed the figure quality using the PLOS-ONE journal ‘PAGE’ tool.

2. Investigate the potential benefits of using NLP-based embeddings and additional feature descriptors beyond those employed in previous studies. Experiment with various feature combinations to optimize performance.

Author’s response: We updated the development process by incorporating eight additional feature descriptors, including the NLP-based Word2Vec. Subsequently, we selected the optimal baseline models for the final model construction. (Please see Word2Vec subsection of Encoding and feature extraction strategy section. See Baseline models: construction and evaluation in the Results section)

3. Explore diverse classifier architectures beyond those used in previous works. Consider incorporating new algorithms and comparing their effectiveness.

Author’s response: As per your suggestion, we utilized additional classifiers (CBC, NB, KNN, LR) that were not employed in the original manuscript. We demonstrated the effectiveness of the additional methods in predicting 2-OM sites. Please see CBC, NB, and KNN subsection of Materials and methods. Furthermore, we compared the proposed method with the existing methods in independent dataset in the subsection of Comparison of Meta-2OM with existing methods. 

4. Provide details on the parameter search ranges used during model development. Include this information in a dedicated README file for transparency and reproducibility.

Author’s response: Thanks for your suggestion. We have included a table in the supplementary materials that details the search ranges for hyperparameters across all machine-learning methods. Additionally, we have added all these information in the README file. (See S1 Table)

5. Utilize all relevant information from baseline models, including probabilistic scores, class labels, and their combinations. Explore different ways of incorporating this information to enhance prediction performance.

Author’s response: Thanks for your suggestion. Initially, we explored all the approaches mentioned and found that features based on probabilistic scores demonstrated a superior discriminative ability. Hence, we developed the final meta-predictor using LR. In the revised manuscript, we also explored six different classifiers to evaluate their effectiveness in meta-prediction. Our results indicate that the LR-based meta-predictor is particularly robust. These details are included in the revised manuscript (Please see the subsection of Impact of meta-learning on the development of Meta-2OM). 

6. Use acronyms consistently throughout the manuscript (e.g., always use ML or Machine Learning, ACC or Accuracy). Define acronyms upon first use and avoid unnecessary acronyms.

Author’s response: Thanks for pointing out the above issue. We have carefully revised the manuscript to ensure consistent use of acronyms throughout. We revised the manuscript according to your comments. (Main Manuscript) 

7. Explain the reasoning behind using a CD-HIT threshold of 0.8. Provide justification for this specific value and its impact on the results.

Author’s response: At this time, we revised the datasets and updated the manuscript accordingly. The main dataset was collected from the latest methods i2OM, where they constructed the datasets by applying CD-HIT analysis with a threshold of 0.8 to control the redundancy. CD-HIT (0.8) indicates the sequence datasets have less than 80% similarity among the sample units. The researcher tried to make a less homogeneous sample for training the model where the datasets can provide unique information to characterize the class. Also, a good-quality dataset can contribute to the training process by producing consistent prediction performance. Due to the lack of biological data samples and considering the overfitting issue, researchers mostly considered an 80% or 90% threshold for developing datasets. (Please see the subsection of Benchmark datasets construction of Methods). 

8. For the user concerns to provide a clear background of training models, authors are advised to cite the recent predictors such as AIPs-SnTCN, iACP-GAEnsC, iACP-GAEnsC, pAtbp-EnC, and cACP-DeepGram. 

Author’s response: We certainly agree with the reviewer. We have cited these methods to provide a clear background of model training and the importance of ensemble learning.

9. Change the title to "Meta-2OM" for consistency and clarity.

Author’s response: We revised and updated the title according to your suggestion.

(Title and Main Manuscript) 

Reviewer #2

This study presents promising Meta2-OM predictors for identifying 2OM sites using meta-learning techniques. The careful dataset construction and exploration of diverse ML classifiers and encodings suggest a well-considered approach. Given the growing interest in RNA post-transcriptional modification sites within bioinformatics, this work holds significant potential. To further strengthen its impact, I have the following suggestions:

1. The authors should investigate both nucleotide-specific and generic models. Compare their advantages and disadvantages, highlighting the strengths of each approach and justifying the choice of using a generic dataset for this study.

Author’s response: As per your suggestion, we have constructed both nucleotide-specific datasets and generic dataset models. The performance comparison showed that the generic model has an advantage over nucleotide-specific models. Therefore, we have selected the generic model. Moreover, the advantages and strengths of the generic model are highlighted in the revised manuscript (Please see the subsection of Comparison of performance based on generic and nucleotide-specific modifications).

2. Clearly explain how the current dataset differs from the one used in i2OM. This will provide context and allow for a more meaningful comparison of the two methods.

Author’s response: Our original dataset shared 66% similarity with those used by the existing methods, making comparisons unreliable and meaningless. Following your suggestion, we significantly revised the manuscript. We now used the same dataset as i2OM and H2Opred, enabling a direct and meaningful assessment of our method’s performance (Please see Benchmark dataset subsection).

3. Include H2Opred in the comparison table alongside other existing methods. Analyze its performance relative to your proposed tool to provide a more comprehensive evaluation.

Author’s response: As per your suggestion, we have compared the performance of the proposed method with H2Opred (Please see Figure 6 and Table 5). 

4. Comparison of Meta2-OM with existing methods. I have noticed they have used just only one method. It should be revised.

Author’s response: In the revised manuscript, we have used two existing methods, which are the most recent. We have provided the reason for excluding several existing methods in the revised manuscript (Please see Comparison of Meta-2OM with existing methods subsection, pages 27-28).

5. Revise Table 5 to ensure a fair comparison with i2OM. If relevant, consider incorporating nucleotide-specific models from both methods as suggested in point 1. 

Author’s response: As per your suggestion, we have revised Table 5 to directly compare our proposed method and existing methods based on the nucleotide-specific datasets (Please see Table 5, page 47). We revised the comparison results as shown in Table 5, where the results were compared based on nucleotide-specific cases for i2OM and H2Opred methods. (Please see Comparison of Meta-2OM with existing methods of the Results subsection, Fig 6 and Table 5).

6. In the machine learning section, the authors should incorporate the following predictors such as iMethyl-STTNC, DP-BINDER, and AFP-CMBPred to provide clear understanding of the model. 

Author’s response: Thanks for your suggestion. We have mentioned the above predictors in the method section of the revised manuscript (Please see pages 17-19 and 23-25).

7. Analyze the optimal sequence length for prediction accuracy. Construct baseline models with varying sequence lengths (e.g., 31-201 bp) and compare their performance to identify the most effective length.

Author’s response: Thank you for your excellent suggestion. In the original manuscript, we carried out the preliminary analysis with multiple lengths using random forest. The result showed that a 41 bp fragment achieved the best result. We therefore carried out computational analysis using this fragment. Notably, several other studies also support the effectiveness of a similar fragment size for different RNA modification site prediction (iR5hmcSC, iRNAD, Deepm5C, m5U-GEPred, TS-m6A-DL, PseU-Pred).

---

## [Decision Letter · Decision Letter 1]

30 May 2024

Meta-2OM: A multi-classifier meta-model for the accurate prediction of RNA 2'-O-methylation sites in human RNA

PONE-D-24-02313R1

Dear Dr. Kurata,

We’re pleased to inform you that your manuscript has been judged scientifically suitable for publication and will be formally accepted for publication once it meets all outstanding technical requirements.

Kind regards,

Shahid Akbar, PhD

Academic Editor

PLOS ONE

Additional Editor Comments (optional):

Reviewers' comments:

Reviewer's Responses to Questions

**Comments to the Author**

1. If the authors have adequately addressed your comments raised in a previous round of review and you feel that this manuscript is now acceptable for publication, you may indicate that here to bypass the “Comments to the Author” section, enter your conflict of interest statement in the “Confidential to Editor” section, and submit your "Accept" recommendation.

Reviewer #1: All comments have been addressed

Reviewer #2: All comments have been addressed

2. Is the manuscript technically sound, and do the data support the conclusions?

Reviewer #1: Yes

Reviewer #2: Yes

3. Has the statistical analysis been performed appropriately and rigorously? 

Reviewer #1: Yes

Reviewer #2: Yes

4. Have the authors made all data underlying the findings in their manuscript fully available?

Reviewer #1: Yes

Reviewer #2: Yes

5. Is the manuscript presented in an intelligible fashion and written in standard English?

Reviewer #1: Yes

Reviewer #2: Yes

6. Review Comments to the Author

Reviewer #1: The required comments are successfully incorporated and paper is significantly improved. The paper can be published from my side.

Reviewer #2: The required comments are successfully incorporated and now the paper is significant improved and should be published from my side

7. PLOS authors have the option to publish the peer review history of their article (what does this mean?). If published, this will include your full peer review and any attached files.

Reviewer #1: No

Reviewer #2: No

---

## [Editor Report · Acceptance letter]

17 Jun 2024

PONE-D-24-02313R1 

PLOS ONE

Dear Dr. Kurata, 

I'm pleased to inform you that your manuscript has been deemed suitable for publication in PLOS ONE. Congratulations! Your manuscript is now being handed over to our production team.

Kind regards, 

on behalf of

Dr. Shahid Akbar 

Academic Editor

PLOS ONE